# Functioning in adult patients with idiopathic inflammatory myopathy: Exploring the role of environmental factors using focus groups

I. Armadans-Tremolosa[1], G. Guilera[2,3], M. Las Heras[1], A. Castrechini[1], A. Selva-O'Callaghan[4]*

1 Department of Social Psychology and Quantitative Psychology, PsicoSAO-Research Group in Social, Environmental, and Organizational Psychology, Universitat de Barcelona, Barcelona, Spain, 2 Department of Social Psychology and Quantitative Psychology, Universitat de Barcelona, Barcelona, Spain, 3 Institute of Neurosciences, Universitat de Barcelona, Barcelona, Spain, 4 Systemic Autoimmune Diseases Unit, Department of Medicine, Vall d'Hebron General Hospital, Universitat Autonoma de Barcelona, Barcelona, Spain

* aselva@vhebron.net

## Abstract

### Objective

Health-related quality of life is impaired in idiopathic inflammatory myopathies. This study aimed to identify the main areas of the health-related quality of life environment domain that are affected in patients with myositis.

### Methods

A qualitative study was performed using focus groups and applying the International Classification of Functioning, Disability, and Health. Participants were recruited from a cohort of 323 adult inflammatory myopathy patients consulting at a reference center for idiopathic inflammatory myopathy in Spain, selected by the maximum variation strategy, and placed in focus groups with 5 to 7 patients per group. The number of focus groups required was determined by data saturation.

### Results

Twenty-five patients distributed in 4 focus groups were interviewed. The verbatim provided 54 categories directly related with environmental factors. Those associated with *products or substances for personal consumption* (e110), *health professionals* (e355), *health services, systems and policies* (e580), *products and technology for personal use in daily living* (e115), and *immediate family* (e310) were the ones most frequently reported.

### Conclusion

The results of this study led to identification of several environmental factors that affect the health-related quality of life of patients with myositis. Remedial interventions should be designed to address some of these factors.

**Data Availability Statement:** All relevant data are within the manuscript and its Supporting information files.

**Funding:** This work was supported in part by the Instituto de Salud Carlos III and the European Regional Development Fund (ERDF) (grant number PI15/02100).

**Competing interests:** The authors have declared that no competing interests exist.

## Introduction

Idiopathic inflammatory myopathies (IIM) are a heterogeneous group of systemic and autoimmune disorders characterized by inflammatory infiltrates on muscle biopsy. Five IIM phenotypes are recognized (dermatomyositis, polymyositis, antisynthetase syndrome, immune-mediated necrotizing myopathy, and sporadic inclusion body myositis), and all five are considered chronic conditions without a cure [1]. Because these disorders affect the musculoskeletal system, an acknowledged problem in myositis patients is difficulties coping with the environment.

Several studies have reported impaired health-related quality of life (HRQoL) in IIM patients [2, 3], and it is considered even worse than the situation in other systemic autoimmune diseases such as systemic lupus erythematous, rheumatoid arthritis, or systemic sclerosis [4]. However, as researchers do not always focus on the same HRQoL domains (ie, physical, mental, social, or environmental), the results obtained are heterogeneous. Data based on large registries have shown that certain clinical characteristics, such as lung disease and joint involvement, are predictors of reduced HRQoL [5]. In addition, other factors, such as severe skin involvement [6], low muscle density detected by computed tomography [7], and certain therapies such as intravenous immunoglobulin [8], have been related with changes in HRQoL. In a previous research, our group established that the HRQoL physical and environmental domains were particularly affected in myositis patients and significantly related with muscle weakness and disease activity [9].

The International Classification of Functioning, Disability and Health (ICF) [10], provides a framework for describing health and the components of well-being. As a classification, it offers a list of environmental factors that interact with a person's health condition. These include features of the physical, social, and attitudinal environment in which people live and conduct their lives, and which can act as facilitators or barriers to a favorable quality of life. The whole category system can be consulted online at https://apps.who.int/classifications/icfbrowser/.

The main aim of this study is focused on how environmental factors affect the daily life of patients with IIM. Based on the ICF classification system, the objectives were to identify, using a focus group format, the main environment-related areas that are affected in this patient population. An understanding of these factors as perceived by the patients is essential to design future programs, activities, or interventions to improve their quality of life.

## Materials and methods

A qualitative study including persons with IIM was conducted using focus groups and applying the International Classification of Functioning, Disability, and Health (ICF) [10] both as a conceptual framework and as a tool for analysis. The ICF is based on an integrative bio-psycho-social model for describing functioning, disability, and health in any health condition. In the ICF framework, problems associated with a health condition can concern *Body functions (b)*, *Body structures (s)*, and *Activities and participation (d)* in a person's life situation. Moreover, all these problems are also influenced by contextual factors such as *Environmental factors (e)* and *Personal factors (pf)*. As a classification system, the ICF provides a list of +1400 indicators (termed ICF categories) hierarchically organized and identified by alphanumeric codes. The letters b, s, d, e, and pf, refer to the components of the classification, and they are followed by numeric codes to identify chapters and categories within chapters. An example of a category from chapter 3 (*Support and relationships*) of the *Environmental factors* component is e355 Health professionals. As a tool for analysis, the ICF was used as a classification system in order

to identify and quantify the various environmental factors that emerged during completion of the focus groups.

The study was approved by the Ethics Commission of Hospital Vall d'Hebron, Barcelona (Spain) (PR (AG) 223/2013), and written informed consent was obtained from each participant. It was performed in accordance with the Declaration of Helsinki. We followed the STROBE [11] statement to improve the quality of reporting of observational studies.

## Participants

Patients were eligible for the study based on the following criteria: 1) diagnosis of probable or definite IIM according to the Bohan and Peter criteria [12, 13] or a score of at least 55% (probable IIM) calculated using the International Myositis Classification Criteria [14]; 2) age at least 18 years; and 3) able to understand the purpose and procedures of the study. Patients were excluded if they had been hospitalized for extremely severe disease or declined participation. To avoid bias, patients with sporadic inclusion body myositis were not included because this phenotype is underrepresented in our myositis outpatient clinic and is, to some extent, a different phenotype (ie, patients do not respond to immunosuppressive therapy).

A random sample of patients from our cohort of 323 adult IIM patients attended in our outpatient clinic (Systemic Autoimmune Diseases Unit of Vall d'Hebron General Hospital, Barcelona, Spain) at any time since 2014 were contacted by telephone and invited to participate in the study. Vall d'Hebron General Hospital is a 700-bed referral and teaching hospital for a catchment population of nearly 400,000 inhabitants. Virtually all patients from the area with suspected myositis are referred to Vall d'Hebron, where they are diagnosed, treated, and followed up, whether or not the disease is severe. Specifically, 38 patients were contacted and 25 agreed to participate (response rate of 66%). All of them fulfilled the previously established inclusion and exclusion criteria. Patients declined to participate mainly because of personal issues (eg, family conciliation or schedule conflicts).

The participants' sociodemographic and clinical data were collected from hospital charts and from our dedicated myositis database. Participants were then selected by the maximum variation strategy [15, 16] based on sex, age, and job situation for distribution into focus groups with 5 to 7 patients per group. The technique used consists in selecting a small number of participants based on the criterion of maximum heterogeneity, in order to provide information about the impact of the disease on different patient profiles [17]. The number of focus groups required was determined by data saturation [18], which yielded a total of 4 groups.

## Data collection

Focus groups were conducted between October 2016 and January 2019. As no relevant advances in myositis treatment occurred during this period, it is not likely that the time interval between the first and the last analysis would have an effect on the results. All focus group sessions were conducted at our hospital Systemic Autoimmune Diseases Unit. Sessions were moderated by a researcher (IAT) trained in conducting group processes and assisted by a master student (MLH) who took field notes. At the beginning of each focus group, participants were again informed of the purpose of the study and the development of the session, and were asked to give their consent to audio record the session.

Focus groups were performed according to specific guidelines for this purpose [19], which included planning, designing, and conducting the focus groups. Sessions were guided by open-ended questions in which participants were asked to discuss their physical and everyday life problems, and the environmental factors or living conditions that acted as facilitators or barriers in their daily functioning. The questions dealt with problems related to three areas: 1)

daily routine (eg, *During a regular day of your daily routine, what are your usual activities*? *What are the main facilitators or barriers that you face in these activities*?); 2) leisure (eg, *What facilitators or barriers do you face in your leisure activities*?); and 3) work activity (eg, *What difficulties or facilities do you experience when you perform productive or work activities*?). The session closed by asking: *What aspects would you like to have to improve your quality of life*? At the end of each session, a summary of the main results was returned to the group to enable participants to verify or amend emergent issues. The duration of the focus groups varied from 1.5 to 2 hours. Sessions were audio recorded, transcribed verbatim, and then analyzed using qualitative content analysis.

Due to the open format of the question script, the environment-related areas were directly elicited by the participants, under the assumption that the concerns and positive points people bring up are those that are relevant to them. Because of this design, the study was unable to capture the importance of each separate environmental factor on the participants' HRQoL, although this could be a goal for future studies.

## Data analysis

**Extraction of concepts and linking to ICF categories.** The procedure of concept extraction and linkage to the ICF categories involved five steps. In the first, the transcription of the discussion generated in each focus group was read through to get an overview of the data collected. Second, data were divided into meaning units, defined as a few consecutive words or sentences with a common theme. In the third step, the concepts contained in each meaning unit were identified, taking into consideration that a meaning unit could contain more than one concept. Fourth, the concepts identified were linked to ICF categories based on established linking rules [20–22] that enable systematic and standardized linkage. An example of concept extraction and linkage to ICF categories is provided in Table 1. All concepts were linked by two independent coders to the most fitting second-level ICF category and, when appropriate, were linked to not definable (nd), not covered (nc), or personal factors (pf). The degree of agreement between the two coders regarding the linked ICF categories was obtained by calculating the Kappa coefficient [23] with 95% bootstrapped confidence intervals (95% CI) using SPSS version 21; the Kappa was interpreted following the Landis and Koch criteria [24]. In the event of disagreement, a third coder was involved in reaching the final decision. Fifth, categories related to *Environmental factors* were selected, as they were the focus of interest of the present study. The environmental factors were analyzed with respect to whether they were facilitators or barriers to functioning.

**Data saturation.** The data saturation was analyzed retrospectively. Data saturation refers to the point at which the investigator has obtained sufficient information from the field. In this study, saturation was reached when the linking process of two consecutive focus groups each

**Table 1. Concept extraction from verbatim and linkage to the ICF: An example of the *Environmental factors* component.**

| Participant | Verbatim | Meaning unit | ICF category |
|---|---|---|---|
| *Moderator* | *Among the main barriers that you face in your daily life, what would you highlight?* | | |
| Patient A | The main problem is not being able to open jars that are tightly closed, the rest, for now, I can manage, thank God. | I can't open jars | e115 Products and technology for personal use in daily living |
| Patient B | I can't go up a step. When I try to go up a step, I fall down | I fall down when I go up a step | e150 Design, construction and building products and technology of buildings for public use |
| Patient C | Yes, I have [problems], but I've been coming anyway, I have to take the subway and all that. | I have to take the subway | e540 Transportation services, systems and policies |

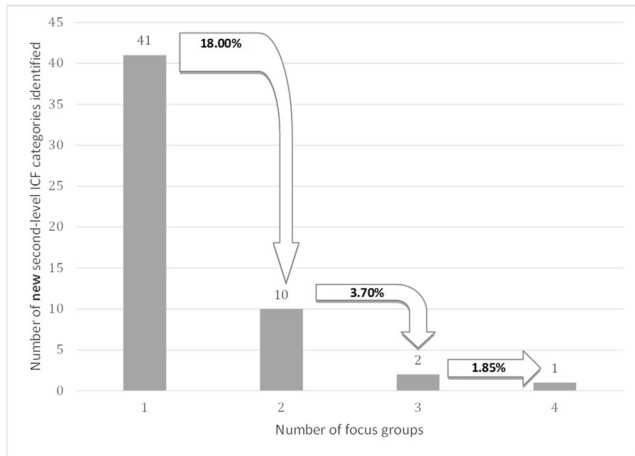

**Fig 1. Saturation of second-level ICF environmental factor categories, from focus groups 1 through 4.** The percentage shows additional categories identified.

reached less than 5% of new second-level ICF categories of the *Environmental factors* categories compared with the number of second-level categories in the previous focus group [25, 26]. As is shown in Fig 1, saturation of data was reached after conducting four focus groups.

## Results

Twenty-five adult IIM patients with a median age of 55.52 years (SD 14.77) participated in the study. The participants' sociodemographic and clinical data are summarized in Table 2.

In total, 2156 concepts were identified in the qualitative analysis of the four transcribed focus group sessions. In all, 171 second-level ICF categories were derived from the text analyses. Of these, 54 categories were directly related to *Environmental factors*, which were distributed into chapters (Table 3). The Kappa coefficient of agreement between the two health professionals in the linking process was 0.67 (95% CI: 0.64–0.70), a value indicating substantial agreement [24].

### Environmental factors

The environmental factors reported most often were those related to *products and technology* (e1), *support and relationships* (e3), and *services, systems and policies* (e5) (Table 3). Thus, these factors seemed to be the most important for patients. The 15 most frequently reported environmental factors and whether they were reported as a facilitator or a barrier are shown in Table 4. The five factors most often reported were related to *products or substances for personal consumption (*e110), *health professionals* (e355), *health services, systems and policies* (e580), *immediate family* (e310), and finally, *products and technology for personal use in daily living (*e115). All these factors were reported as both facilitators and barriers. Examples of the environmental factors acting as facilitators or barriers are shown in Table 5.

## Discussion

In the current study, we investigated the environmental factors that have an impact on daily living reported by IIM patients, using focus groups and the ICF as a reference. The ICF categories most often cited by our patients were related to products or substances for personal consumption (e110), such as medication and disease-related healthy food issues in the diet.

**Table 2. Sociodemographic and clinical data of the participants (*n* = 25) [27].**

| Sociodemographic data | Description |
|---|---|
| Age: mean (SD), range | 55.52 (14.77), 21–76 |
| Sex | |
| Female | 18 (72.0) |
| Male | 7 (28.0) |
| Marital status | |
| Married/cohabitant | 18 (72.0) |
| Single | 4 |
| Divorced or widow/er | 3 |
| Living arrangement | |
| Family | 24 (96.0) |
| Alone | 1 (4.0) |
| Educational level | |
| Primary education | 12 (48.0) |
| Secondary education | 8 (32.0) |
| Higher education | 3 (12.0) |
| Other or unknown | 2 (8.0) |
| Employment status | |
| Pensioners | 21 (84.0) |
| Wage earner | 4 (16.0) |
| Clinical data | |
| Diagnosis | |
| Dermatomyositis* | 15 (60.0) |
| Antisynthetase syndrome | 5 (20.0) |
| Immune-mediated necrotizing myopathy | 3 (12.0) |
| Polymyositis | 2 (8.0) |
| Age at diagnosis: mean (SD), range | 48.64 (15.04), 20–74 |
| Years of follow-up: mean (SD), range | 7.54 (8.92), 0.2–39.9 |
| Range of disease activity in all patients Physician Global Assessment of Disease Activity (VAS) | From 0 to 4 |

Data are reported as n (%), unless otherwise indicated.

VAS, visual analogue scale. A 10-cm visual analog scale anchored with the descriptors no activity (0) and maximum activity (10) [27].

*No patients were categorized as having amyopathic dermatomyositis.

**Table 3. Second-level ICF categories by chapters from the *Environmental factors* component and the corresponding frequency of appearance.**

| Chapters | Second-level ICF category (*n*) | Frequency of appearance |
|---|---|---|
| e1. Products and technology | 12 | 82 |
| e3. Support and relationships | 12 | 79 |
| e5. Services, systems and policies | 12 | 70 |
| e4. Attitudes | 10 | 35 |
| e2. Natural environment and human-made changes to environment | 8 | 32 |

**Table 4. Top 15 most frequently reported ICF categories from the *Environmental factors* component, categorized as facilitators or barriers to functioning.**

|  | ICF category title | Frequency of appearance | Facilitator | Barrier |
|---|---|---|---|---|
| e110 | Products or substances for personal consumption | 23 | × | × |
| e355 | Health professionals | 23 | × | × |
| e580 | Health services, systems and policies | 22 | × | × |
| e310 | Immediate family | 20 | × | × |
| e115 | Products and technology for personal use in daily living | 18 | × | × |
| e150 | Design, construction and building products and technology of buildings for public use | 14 | × | × |
| e340 | Personal care providers and personal assistants | 12 | × | n.i. |
| e225 | Climate | 10 | × | × |
| e210 | Physical geography | 9 | × | × |
| e535 | Communication services, systems and policies | 8 | × | × |
| e555 | Associations and organizational services, systems and policies | 8 | × | × |
| e460 | Societal attitudes | 7 | *n.i.* | × |
| e120 | Products and technology for personal indoor and outdoor mobility and transportation | 7 | × | n.i. |
| e560 | Media services, systems and policies | 7 | × | × |
| e590 | Labor and employment services, systems and policies | 6 | × | × |

n.i.: not identified.

Patients also highlighted the importance of the relationship with health professionals (e355), including doctors, nurses, psychologists, and physiotherapists, accessibility to health services (e580), technological support (e115), the role of caregivers (e310), and certain problems related with the disability caused by the disease, such as muscle weakness, intolerance to physical activity, and fatigue, with their consequences on mobility.

Patients with idiopathic inflammatory myopathies can be highly dependent because of muscle weakness, joint inflammation, or the principal organ involvement, affecting, for example, respiration or the swallowing process [28]. Hence, it is essential to identify the factors that act as facilitators or barriers in their lives. Once these are known they can be addressed to improve the patients' clinical and social status, well-being, and HRQoL. The focus-group setting used here was well-accepted by all participants, providing them with a space for interaction and support where they could communicate their experience with the disease.

In addition to the classical approach (drug therapy) for managing these patients, other elements such as adapted stools for the shower, banisters to help going upstairs, and adapted living spaces focusing on building designs, were identified as factors that contribute to aid everyday functioning. As systematic reviews of orthotic devices for patients with neuromuscular problems have failed to find high-quality evidence proving their effectiveness [29], there is a lack of reliable information in this line for physicians and patients. The recent emergence of what may be useful devices, such as exoskeletons for myositis patients [30], could be of help for the problems in category e115 (Products and technology for personal use and daily living) and e120 (Products and technology for personal indoor and outdoor mobility), brought to light by patients in the focus groups. The role of relatives or assistants was also significant, as they help with walking and standing up, and confer reassurance on the patients' ability to walk by avoiding the risk of falls. Thus, the importance of family members as caregivers also emerged in the sessions. The use of patient-reported outcomes derived from focus groups to raise awareness of issues that matter to patients is precisely what enabled identification of these facilitators. Hence, the focus group strategy is a worthwhile approach that should be considered in future studies as other authors have advocated [31].

**Table 5. Examples of the environmental factors acting as facilitators or barriers.**

| ICF code | ICF category title | Example | Classification |
|---|---|---|---|
| e110 | Products or substances for personal consumption | "[…] currently, for example, 11 months ago let's say I started to live a normal life, thanks to **cortisone and immunoglobulin**. And now, when I get up I take cortisone and…" | Facilitator |
| | | "[…] what the doctors do is give you a **medication** that works well for you, because **before getting to that point I tried 40**; all of them were bad for me." | Barrier |
| e355 | Health professionals | "I went to the psychologist and he told me 'Look, write. If it's hard for you to talk to someone, then write', and I had no problem with that. **It also helped me a lot** to get out everything I was holding inside, the feeling of being powerless and all that." | Facilitator |
| | | Well, for that, for example, you go to the GP. In my case, yes, but what happens is that **the GPs confuse you.** | Barrier |
| e580 | Health services, systems and policies | "[…] because of course, **physiotherapy, you have to pay for it, and physiotherapy is very important** because it helps a lot." | Facilitator |
| | | "but I do see […] that there **is a lack of resources so that the appointments are not so long,** the waiting time, etc." | Barrier |
| e310 | Immediate family | "When I go for a walk in the afternoon, **I always go with my husband, he always comes with me,** I always hold on to him." | Facilitator |
| | | "My husband's help? **He's used to me doing everything for him,** always…" | Barrier |
| e115 | Products and technology for personal use in daily living | "I sit on **a stool,** in the shower; **I have everything adapted,** because I couldn't even get in; **handrails** everywhere for the shower." | Facilitator |
| | | "…but these days, except this: **I can't open jars that are tightly closed."** | Barrier |
| e150 | Design, construction and building products and technology of buildings for public use | "[…] **if there's an elevator, I take the elevator**. If not, if there are stairs up, well I have to take it very calmly, because it's hard. If not, I get fatigued." | Facilitator |
| | | **"I can't go up stairs,** I can't walk." | Barrier |
| e340 | Personal care providers and personal assistants | "[…] the last two years, I'm getting a little better, less pain; before, I'm in pain 24 hours. When I'm sleepy, I can't get up, for example. **I might need someone to hold on to.** | Facilitator |
| e225 | Climate | "because in **summer** it's a happy time and you go out, etc." | Facilitator |
| | | "[…] **but in winter, you have to stay at home more,** or go out less because of the **rain** or the **cold**, and I think that being like that, so still, it's like the pain is more concentrated". | Barrier |
| e210 | Physical geography | "For example, when I go out for a walk and I go to the mountains, I'm another person. **I feel much better.**" | Facilitator |
| | | "Yes, my problem was going uphill., **I live on a hill,** so what happened? Of course, **I had a really hard time going up a whole way** and I had to stop half way up the street more than once." | Barrier |
| e535 | Communication services, systems and policies | "You have an enormous lack of knowledge and **on the internet they at least explain something.**" | Facilitator |
| | | "I think many things are missing; people working on these rare diseases. **There is a lack of information."** | Barrier |
| e555 | Associations and organizational services, systems and policies | "[…] if they put me in a group like an **association,** those that support ill people, well I think that would be good…." | Facilitator |
| | | "We've looked for groups, we've looked with my wife, with my family, and you find groups. There's an association for this cancer, for that cancer, for stroke, for this and that, **but there is no association for dermatomyositis.**" | Barrier |
| e460 | Societal attitudes | **"But no one believes you're sick.** That's another thing that affects me at least, isn't it? | Barrier |
| e120 | Products and technology for personal indoor and outdoor mobility and transportation | "[…] I fell down on the stairs. I started to walk again, with a **walker**" | Facilitator |
| e560 | Media services, systems and policies | "[…] I have always looked at the disease **online (internet)**, I like to find out everything." | Facilitator |
| | | "But, of course, I don't know if the thing is good or bad. **There is a lack of information about this.**" | Barrier |
| e590 | Labor and employment services, systems and policies | **"I had to ask for it [a leave of absence from work]. I was almost a year away.** And when I felt better I told the doctor: 'I need to go back to my routine, because I have to see if I can or not'." | Facilitator |
| | | "How is it possible that a person with so much knowledge **has to go on unemployment** or has to work doing…oh well, whatever." | Barrier |

Lack of acknowledgment of their disease by other people and society in general (e460 Societal attitudes) was also a recognized environmental factor affecting the well-being of IIM patients. Patients need more useful lay information and dedicated associations where solutions can be sought for their self-perceived problems, similar to those already functioning around the world (eg, The Myositis Association, in the United States) (e555 Associations and organizational services, systems, and policies). It is well recognized that reliable information about the disease is crucial for IIM patients (e535 Communication), and this point is suboptimal in our setting. Contact e-mails for health professionals, webs (internet) (e560 Media Services, systems and policies), and handouts, fliers, or patient page leaflets focused on IIM should be developed to improve access to this information [32]. Web meetings and telematic symposiums should be conducted to empower patients to gain knowledge about their disease, the available therapeutic options, and the environmental modifications that will improve their well-being and HRQoL.

The main limitation of the present study is that all patients evaluated were attended in an outpatient clinic, which effectively omitted those in a more severe clinical situation. Moreover, the single center design, and the fact that most participants were pensioners, are issues that preclude generalization of the results to the overall population of patients with idiopathic inflammatory myopathy. In contrast, use of the ICF as a reference framework and the expertise of the team who performed the focus groups and linking process are strengths of the study.

In conclusion, the ICF categories related to environmental factors identified here are of value for clinicians treating patients with these rare diseases, as they raise awareness of the need to reinforce the facilitating role of specific environmental factors and reduce the negative impact of others to improve the patients' daily functioning.

## Supporting information

**S1 Data.**
(XLSX)

## Acknowledgments

We thank Ms. Celine Cavallo for English language support.

## Author Contributions

**Conceptualization:** I. Armadans-Tremolosa, G. Guilera, M. Las Heras, A. Castrechini, A. Selva-O'Callaghan.

**Data curation:** I. Armadans-Tremolosa, G. Guilera, M. Las Heras, A. Castrechini, A. Selva-O'Callaghan.

**Formal analysis:** I. Armadans-Tremolosa, G. Guilera, M. Las Heras, A. Castrechini, A. Selva-O'Callaghan.

**Funding acquisition:** I. Armadans-Tremolosa, G. Guilera, A. Castrechini, A. Selva-O'Callaghan.

**Investigation:** I. Armadans-Tremolosa, G. Guilera, M. Las Heras, A. Castrechini, A. Selva-O'Callaghan.

**Methodology:** I. Armadans-Tremolosa, G. Guilera, M. Las Heras, A. Castrechini, A. Selva-O'Callaghan.

**Writing – review & editing:** I. Armadans-Tremolosa, G. Guilera, M. Las Heras, A. Castrechini, A. Selva-O'Callaghan.

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
