## [Decision Letter · Decision Letter 0]

4 Aug 2020

PONE-D-20-10513

“Functioning in adult patients with idiopathic inflammatory myopathy: Exploring the role of environmental factors using focus groups”.

PLOS ONE

Dear Dr. Selva-O'Callaghan,

Thank you for submitting your manuscript to PLOS ONE. After careful consideration, we feel that it has merit but does not fully meet PLOS ONE’s publication criteria as it currently stands. Therefore, we invite you to submit a revised version of the manuscript that addresses the points raised during the review process.

Thank you for the opportunity to review this submission to PLOS One. The manuscript, as the reviewers themselves agree, is interesting and well-conceptualised. As such, I would encourage the authors to submit a revision of their manuscript for consideration for publication in PLOS One.

In addition to the points raised by the reviewers, I hope that the authors would also take into consideration the following comments and suggestions for their revised manuscript:

1. More information is required, or perhaps needs to be explicated, in the methods section:

(a) what is the total sampling frame from which participants were recruited (seems like this is n;

(b) how many of these identified participants fit the inclusion and exclusion criteria (seems like this is n = 38);

(c) of the identified participants, how many agreed to participate (seems like this is n = 25); (d) what is the response rate; and (e) what are the reasons for non-participation, if any?

2. Given the multidisciplinary nature of this journal, please also discuss very briefly in the methods:

(a) the delay between the commencement of recruitment (2014) and focus group discussion/data collection (Oct 2016 - Jan 2019), and how this may (or does not) influence findings (please provide context especially if these findings are likely to have remained ostensibly stable across the time periods; i.e., no significant change in healthcare policy or medication advancements etc.);

(b) the justification behind maximum variation sampling strategy employed for this study, especially given there are some minority/socially disadvantaged groups that may have not felt comfortable providing individual information in a focus-group setting; and

(c) why thematic analyses (vs. other analytic frameworks) was employed or particularly suitable for this study.

3. It would be helpful for readers if the interview guide (i.e., Table 1) was just provided in-text. It would also be helpful if the authors could briefly comment on whether the study managed to capture the importance of each environmental factor on the QOL of participants; if not, whether there is any value in doing this in future studies, etc.

4. Please also state the software used to derive the coefficients, or used in the analyses of data, if any. Please also discuss whether the kappa coefficient is appropriate, or if this is representative of the varied semantic influence of the factors upon the observer.

5. Please comment on how such a classification for analysing and understanding the data may be somewhat limiting (or is not limiting at all). It may thus also be helpful to identify the frequencies of units identified as either a facilitator, or barrier, or both, such that a more nuanced understanding of the data can be appreciated. It would also be helpful to know which other environmental factors (deemed nd or nc by coders, if any) came up such that it may potentially help facilitate the development and understanding of factors influencing IIM; otherwise, it would also be helpful for readers should the authors detail the holistic and comprehensive nature of the ICF coding.

We look forward to receiving your revised manuscript.

Kind regards,

Haikel A. Lim, MD, MSc

Academic Editor

PLOS ONE

Journal Requirements:

2. Please include your tables as part of your main manuscript and remove the individual files. Please note that supplementary tables (should remain/ be uploaded) as separate "supporting information" files

Reviewers' comments:

Reviewer's Responses to Questions

**Comments to the Author**

1. Is the manuscript technically sound, and do the data support the conclusions?

Reviewer #1: Yes

Reviewer #2: Yes

Reviewer #3: Yes

2. Has the statistical analysis been performed appropriately and rigorously? 

Reviewer #1: N/A

Reviewer #2: Yes

Reviewer #3: Yes

3. Have the authors made all data underlying the findings in their manuscript fully available?

Reviewer #1: Yes

Reviewer #2: Yes

Reviewer #3: Yes

4. Is the manuscript presented in an intelligible fashion and written in standard English?

Reviewer #1: Yes

Reviewer #2: Yes

Reviewer #3: Yes

5. Review Comments to the Author

Reviewer #1: This manuscript is extremely well written to an extent i can't find anything to critique. Some minor points:

1. Pls insert the table into the main text rather than suppl

Thank you for the opportunity to review.

Reviewer #2: This study aims to identify the self-reported environmental factors which significantly impair the health-related quality of life of adult patients with idiopathic inflammatory myopathy through focused group discussions. The 3 most commonly reported environmental factors identified are: Products or substances for personal consumption, Health professionals, and Health services, systems and policies. These will help to guide measures to improve the quality of life in these patients. The manuscript is clear and concise.

Major points

1. What was the language medium of the focus group discussion? If a different language from English was used, please clarify how an accurate translation was achieved.

2. Please clarify what “focus group guidelines” encompass (page 6, paragraph 3).

3. In Table 1, question 3 asks “When you perform productive or work activities, what difficulties and/or advantages do you experience?”. Please clarify what exactly “advantages” refer to.

4. When an appropriate second-level ICF category is not available for linking to concepts, they may be linked to “not definable (nd)” or “not covered (nc)”. These unclassified concepts may remain important even though they do not fall within the ICF framework. Were any such concepts relatable to the environment? If any, how important are they?

5. Please provide clinical data representing the disease activity status of the patients in Table III as disease activity may influence the relative impact that environmental factors have on quality of life as compared to other factors (for example, disease factors and personal factors).

6. Please include in Supplementary Table S1 examples of how each environmental factor may act as a facilitator and a barrier, if it has been found to do so as recorded in Table V.

7. Please clarify what is meant by “certain problems related with the disability caused by the disease” in the discussion (page 9, paragraph 1).

8. In the discussion, the sentence “The use of patient-reported outcomes derived from the focus group strategy to raise awareness of the issues that matter to patients is an essential part that should be considered in future studies (26).” (page 10, paragraph 2) does not appear to be related to the rest of the paragraph which discusses the role of caregivers. Please rephrase or move this statement to its own paragraph.

9. Please shorten the concluding paragraph and move new suggestions to the main discussion text.

Minor points

1. Please standardize in-line citation formatting throughout the manuscript (e.g., citation of reference 10 on page 4, paragraph 3).

2. Please remove the “a” in the following sentence “Sporadic inclusion body myositis was... and a to some extent…” (page 5, paragraph 3).

3. Please remove the opening sentence of the “Environmental factors” subsection of “Results” (page 8, paragraph 3) as the definition of environmental factors has already been provided in the “Introduction”.

4. Reference 23 (Opinc AH, Brzezińska OE, Makowska JS. Disability in idiopathic inflammatory myopathies: questionnaire-based study. Rheumatol Int. 2019; 39:1213-1220.) does not appear to be cited within the manuscript. Please clarify its role in the text.

5. Reference 27 (Punnoose AR, Burke AE, Golub RM. JAMA patient page. Muscular dystrophy. JAMA. 2011; 306:2526.) is a patient information sheet for “Muscular dystrophy”. Please use a more relevant reference (for example, the Patient Fact Sheet on inflammatory myopathies as published by the American College of Rheumatology at https://www.rheumatology.org/Portals/0/Files/Inflammatory-Myopathies-Fact-Sheet.pdf).

Reviewer #3: Well-written study that gives an insightful understanding into the challenges faced by a group of patients with an uncommon rheumatic disease (IIM). Excellent use of an organized ICF framework, focused group, and well-conducted qualitative/thematic analysis of the environmental factors (that act as facilitators or barriers for this group of patients) by 2-3 independent persons as coders of meaning units. The study fulfils all field of the publication criteria including good demonstration of rigorous statistical analysis in qualitative studies. Good discussion points, including a balanced recognition of important limitations (eg single centre & hence caution with generalisability of the results) and strengths. Your study inspires and gives a great model that other centres around the world (of different social and cultural make-up) can adopt to study the challenges that their patients with IIM/other rheumatic diseases face.

Minor grammatical errors (2 of which stood out to me) did not affect the overall good quality of this manuscript.

6. PLOS authors have the option to publish the peer review history of their article (what does this mean?). If published, this will include your full peer review and any attached files.

Reviewer #1: No

Reviewer #2: No

Reviewer #3: **Yes: **Stanley Angkodjojo

---

## [Author Response · Author response to Decision Letter 0]

3 Oct 2020

PONE-D-20-10513

“Functioning in adult patients with idiopathic inflammatory myopathy: Exploring the role of environmental factors using focus groups”.

PLOS ONE

Dear Haikel A. Lim, 

We appreciate the opportunity to resubmit a revised version of our manuscript. We would like to thank you and the reviewers for your comments. Your careful reading of the paper and valuable suggestions and remarks have enabled us to significantly improve our study. 

Detailed responses are given below to each of the academic editor’s and reviewers’ comments. All changes in the manuscript are marked in the file labeled 'Revised Manuscript with Track Changes'.

We hope that in its present form you will find our manuscript suitable for publication.

Kind regards from the research group. 

Academic Editor:

1. More information is required, or perhaps needs to be explicated, in the methods section:

(a) what is the total sampling frame from which participants were recruited (seems like this is n;

(b) how many of these identified participants fit the inclusion and exclusion criteria (seems like this is n = 38);

(c) of the identified participants, how many agreed to participate (seems like this is n = 25); (d) what is the response rate; and (e) what are the reasons for non-participation, if any?

[Response] Thanks for your comment. In the “Participants” section we explained that the sampling frame was the cohort of myositis patients (n = 323) attended in our center in Barcelona. We have included additional information in the text as follows (indicated here in yellow highlight): 

“A random sample of patients from our cohort of 323 adult IIM patients attended in our outpatient clinic (Systemic Autoimmune Diseases Unit of Vall d’Hebron General Hospital, Barcelona, Spain) at any time since 2014 were contacted by telephone and invited to participate in the study. Vall d’Hebron is a 700-bed referral and teaching hospital serving a catchment population of nearly 400,000 inhabitants. Virtually all patients from the area with suspected myositis are referred to Vall d’Hebron, where they are diagnosed, treated, and followed up, whether or not the disease is severe. Specifically, 38 patients were contacted and 25 agreed to participate (response rate of 66%). All of them fulfilled the previously established inclusion and exclusion criteria. Patients declined to participate mainly because of personal issues (eg, family conciliation or schedule conflicts).”

2. Given the multidisciplinary nature of this journal, please also discuss very briefly in the methods:

(a) the delay between the commencement of recruitment (2014) and focus group discussion/data collection (Oct 2016 - Jan 2019), and how this may (or does not) influence findings (please provide context especially if these findings are likely to have remained ostensibly stable across the time periods; i.e., no significant change in healthcare policy or medication advancements etc.);

(b) the justification behind maximum variation sampling strategy employed for this study, especially given there are some minority/socially disadvantaged groups that may have not felt comfortable providing individual information in a focus-group setting; and

(c) why thematic analyses (vs. other analytic frameworks) was employed or particularly suitable for this study.

[Response] We appreciate the Editor’s comments to improve our manuscript. 

a) We initiated recruitment in 2014, and focus groups were conducted between 2016 to 2019. The delay mentioned was due to difficulties establishing a shared schedule between the different actors of the study: psychologists, physicians, and patients. Nevertheless, we believe this rather lengthy period would not affect to the results obtained, the issue that seems to concern the Editor. Specific therapy for myositis has not substantially changed over these years. The last Cochrane Database Systematic Review on Myositis was published in 2012 (Gordon PA, Winer JB, Hoogendijk JE, Choy EH. Immunosuppressant and immunomodulatory treatment for dermatomyositis and polymyositis. Cochrane Database Syst Rev. 2012;2012(8): CD003643. Published 2012 Aug 15. doi: 10.1002 /14651858. CD003643.pub4). The most recently published randomized trials focusing on rituximab in myositis and bimagrumab in sporadic inclusion myositis (the latter patients not included in our study), are from 2013 (Oddis CV, Reed AM, Aggarwal R, et al. Rituximab in the treatment of refractory adult and juvenile dermatomyositis and adult polymyositis: a randomized, placebo-phase trial. Arthritis Rheum. 2013;65(2):314-324. doi:10.1002/art.37754) and 2019 (Hanna MG, Badrising UA, Benveniste O, et al. Safety and efficacy of intravenous bimagrumab in inclusion body myositis (RESILIENT): a randomized, double-blind, placebo-controlled phase 2b trial. Lancet Neurol. 2019;18(9):834-844. doi:10.1016/S1474-4422(19)30200-5), respectively. Therefore it is unlikely that new drug treatments would affect in the outcomes in patients analyzed from 2016 to 2019.

For the sake of the clarity, we have added a paragraph as follows (in yellow):

“Focus groups were conducted between October 2016 and January 2019. As no relevant advances in myositis treatment occurred during this period, it is unlikely that the time interval between the first and the last analysis would have an effect on the results. All focus group sessions were conducted at our hospital Systemic Autoimmune Diseases Unit. Sessions were moderated by a researcher (IAT) trained in conducting group processes and assisted by a master student (MLH) who took field notes.”

b) We have added a sentence to Materials and methods regarding the maximum variation sampling strategy as follows (in yellow here, on page 6 in red):

“Participants were then selected by the maximum variation strategy (15, 16) based on sex, age, and job situation for distribution into focus groups with 5 to 7 patients per group. The technique used consists in selecting a small number of participants based on the criterion of maximum heterogeneity, in order to provide information about the impact of the disease on patients with different profiles (new reference (17) Palinkas et al. 2015). “

As to the comment about certain groups feeling uncomfortable in the focus group setting, in our experience this was not the case with the participants included and we have added this sentence to the discussion: 

“The focus group setting was well-accepted by participants, providing them with a space for interaction and support where they could communicate their experience with the disease.”

c) Thank you for bringing up this issue. The boundaries between qualitative content analysis and thematic analysis have not been clearly specified, and both are commonly used analytic approaches for large pieces of text (ref: Vaismoradi, M., Turunen, H., & Bondas, T. (2013). Content analysis and thematic analysis: Implications for conducting a qualitative descriptive study. Nursing & Health Sciences, 15(3), 398-405.), such as those collected through interviews or focus groups. Qualitative content analysis and thematic analysis would both have been suitable for analyzing the type of data we collected in our study. The ICF Research Branch, as a collaborating center of the World Health Organization, in the ICF linking studies, strongly recommends not only to identify themes within a set of texts by using the well-established linking rules, but also to quantify the appearance of themes, and these recommendations better fit the content analysis approach. Most studies carried out in the ICF framework and using interviews or focus groups apply a qualitative content analysis (Coenen M, Cieza A, Stamm TA, Amann E, Kollerits B, Stucki G. Validation of the International Classification of Functioning, Disability and Health (ICF) Core Set for rheumatoid arthritis from the patient perspective using focus groups. Arthritis Res Ther. 2006;8(4):R84; Meesters J, Pont W, Beaart-Van De Voorde L, Stamm T, Vliet Vlieland T. Do rehabilitation tools cover the perspective of patients with rheumatoid arthritis? A focus group study using the ICF as a reference. Eur J Phys Rehabil Med. 2014;50(2):171-184.). In this line, we believe our study is more aligned to qualitative content analysis than thematic analysis. We have specified the use of qualitative content analysis the text, as follows:

“Sessions were audio recorded, transcribed verbatim, and analyzed using qualitative content analysis.” 

3. It would be helpful for readers if the interview guide (i.e., Table 1) was just provided in-text. It would also be helpful if the authors could briefly comment on whether the study managed to capture the importance of each environmental factor on the QOL of participants; if not, whether there is any value in doing this in future studies, etc.

[Response] Following the suggestion, the interview guide has been eliminated as a Table and provided in-text as follows:

“Focus groups were performed according to focus group guidelines. Sessions were guided by open-ended questions in which participants were asked to discuss their physical and everyday life problems, and the environmental factors or living conditions that acted as facilitators or barriers in their daily functioning. The questions dealt with problems related to three areas: 1) daily routine (eg, During a regular day of your daily routine, what are your usual activities? What are the main facilitators or barriers that you face in these activities?); 2) leisure (eg, What facilitators or barriers do you face in your leisure activities?); and 3) work activity (eg, What difficulties or benefits do you experience when you perform productive or work activities?). The session closed by asking: What aspects would you like to have to improve your quality of life? 

We have added this comment to explain that the effect of each separate environmental factor could not be determined: 

Due to the open format of the question script, the environment-related areas were directly elicited by the participants, under the assumption that the concerns and positive points people bring up are those that are relevant to them. Because of this design, the study was unable to capture the importance of each separate environmental factor on the participants’ HRQoL, although this could be a goal for future studies.”

4. Please also state the software used to derive the coefficients, or used in the analyses of data, if any. Please also discuss whether the kappa coefficient is appropriate, or if this is representative of the varied semantic influence of the factors upon the observer.

[Response] In keeping with the suggestion, the software used to compute the Kappa coefficient has been specified in Methods. The Kappa was interpreted following the well-established rules of Landis and Koch (1977). Information added as follows:

“All concepts were linked by two independent coders to the most fitting second-level ICF category and, when appropriate, were linked to not definable (nd), not covered (nc), or personal factors (pf). The degree of agreement between the two coders regarding the linked ICF categories was obtained by calculating the Kappa coefficient (23) with 95% bootstrapped confidence intervals (95% CI) using SPSS version 21; the Kappa was interpreted following the Landis and Koch criteria (24).” 

And in Results :

“The Kappa coefficient of agreement between the two health professionals in the linking process was 0.67 (95% CI: 0.64 - 0.70), a value indicating substantial agreement (24).” 

5. Please comment on how such a classification for analysing and understanding the data may be somewhat limiting (or is not limiting at all). It may thus also be helpful to identify the frequencies of units identified as either a facilitator, or barrier, or both, such that a more nuanced understanding of the data can be appreciated. It would also be helpful to know which other environmental factors (deemed nd or nc by coders, if any) came up such that it may potentially help facilitate the development and understanding of factors influencing IIM; otherwise, it would also be helpful for readers should the authors detail the holistic and comprehensive nature of the ICF coding.

[Response] The ICF is a broad, internationally accepted classification system that uses a unified, standard language and framework for describing functioning, disability, and health in any health condition (10), including IIM. The ICF includes more than 1400 categories. Therefore, it is probably one of the most comprehensive category systems currently in existence to describe functioning. In this sense, the use of this system is a strength of the study, as was mentioned in the Discussion. However, as occurs with any classification system, it has its limitations, as does the coding process used to implement it. 

Although the ICF is comprehensive, it does not cover all concepts or is not able to classify some concepts; hence, the categories “not covered” or “not definable” are used in these cases. In the present study, only three concepts were linked to ‘not covered’, and none of them referred to an environmental factor. As to "not definable" concepts, although they were more numerous (n = 39), none of them were related to environmental factors. Instead they referred to aspects that were so broad and generic that they could not be assigned to any specific ICF category (eg, quality of life or recovery). Hence we can say that the ICF was useful in our study, as it enabled classification of all environment-related factors.

In relation to the coding process, two coders were involved to ensure the reliability of the categories extracted. Agreement between the coders was satisfactory and similar to that reported in other studies using the ICF as a classification framework (Sveen U, Ostensjo S, Laxe S, Soberg HL. Problems in functioning after a mild traumatic brain injury within the ICF framework: the patient perspective using focus groups. Disabil Rehabil. 2013;35(9):749-757). 

To our mind, these aspects are already represented in the manuscript and we have not included additional information in that regard. However, if the Editor believes more data is needed, we will be glad to provide it. 

In relation to the facilitating or barrier role of the environmental factors, instead of including the frequency of appearance, we have included examples in Table V for each environmental factor, in line with the request of reviewer 2. In addition, to clarify the idea that we analyzed the environmental factors as facilitators and barriers, we have modified the text as follows:

“Fifth, categories related to Environmental factors were selected, as they are the focus of interest of the present study. The environmental factors were analyzed with respect to whether they were facilitators or barriers to functioning.”

We hope these changes are sufficient and to the Editor’s satisfaction.

Reviewer #1: 

This manuscript is extremely well written to an extent i can't find anything to critique. Thank you for the opportunity to review.

[Response] Thank you for your kind critique.

Some minor points:

1. Pls insert the table into the main text rather than suppl

[Response] Done.

Reviewer #2: 

This study aims to identify the self-reported environmental factors which significantly impair the health-related quality of life of adult patients with idiopathic inflammatory myopathy through focused group discussions. The 3 most commonly reported environmental factors identified are: Products or substances for personal consumption, Health professionals, and Health services, systems and policies. These will help to guide measures to improve the quality of life in these patients. The manuscript is clear and concise.

Major points

1. What was the language medium of the focus group discussion? If a different language from English was used, please clarify how an accurate translation was achieved. 

[Response] The language used during the focus group was Spanish. A native English-speaking language consultant specialized in medical research papers revised the article, which was written in English, and revised the translation of the focus discussion items included (questions and responses). The present text has been checked again to correct minor grammatical errors mentioned by Reviewer 3 and the questions and responses have been revised against the source text in Spanish. We have added an acknowledgement about the language help at the end of the article: “Acknowledgment: We thank Ms. Celine Cavallo for English language support. 

2. Please clarify what “focus group guidelines” encompass (page 6, paragraph 3).

[Response] We have added a sentence (in yellow) and a reference.

“Focus groups were performed according to specific guidelines for this purpose (19), which included planning, designing, and conducting the focus groups. Sessions…”

(ref) Onwuegbuzie AJ, Dickinson WB, Leech NL, Zoran AG. A Qualitative Framework for Collecting and Analyzing Data in Focus Group Research. Int J Qual Methods. 2009; 8:1-21.

3. In Table 1, question 3 asks “When you perform productive or work activities, what difficulties and/or advantages do you experience?”. Please clarify what exactly “advantages” refer to.

[Response] We were referring to “facilities for the patients. To clarify, we have changed the word “advantages” to “facilities”.

4. When an appropriate second-level ICF category is not available for linking to concepts, they may be linked to “not definable (nd)” or “not covered (nc)”. These unclassified concepts may remain important even though they do not fall within the ICF framework. Were any such concepts relatable to the environment? If any, how important are they?

[Response] Only three concepts were linked to ‘not covered’, and none of them referred to environmental factors.

As to "not definable" concepts, although they were more numerous (n = 39), none of them were related to environmental factors. Instead they referred to aspects that were so broad and generic that they could not be assigned to any specific ICF category (eg, quality of life or recovery). We decided not to include this information in the manuscript, but if the reviewer believes it is needed, we will be glad to provide it.

5. Please provide clinical data representing the disease activity status of the patients in Table III as disease activity may influence the relative impact that environmental factors have on quality of life as compared to other factors (for example, disease factors and personal factors).

[Response] We understand the reviewer’s concern. One of the exclusion criteria was hospitalization for severe disease, and all patients were recruited from our outpatient clinic; hence, it is unlikely that there were striking differences in the activity pattern of the disease at the time of the focus group. Nevertheless, we were able to retrieve from our hospital database and charts a tentative activity evaluation measured by the Physician Global Assessment of Disease Activity which uses a 10-cm visual analog scale anchored with the descriptors no activity (0) and maximum activity (ref.). In Table III, we have included an estimation of disease activity, which was recorded to be between 0 and 4.

(ref) International Myositis Assessment and Clinical Studies (IMACS) Group website. https://dirapps.niehs.nih.gov/imacs/. Accessed 12 August 2020

6. Please include in Supplementary Table S1 examples of how each environmental factor may act as a facilitator and a barrier, if it has been found to do so as recorded in Table V.

[Response] Thank you for pointing this out. Examples of how each environmental factor acts as a facilitator or a barrier have been included in the table. We hope it will increase the understanding of both sides of the same environmental factor. Since Reviewer 1 suggested to include Supplementary Table S1 in the main text, this information has been included in Table V as follows (we have modified the title of the table accordingly)

Table V. Examples of the Environmental factors acting as facilitators or barriers

ICF code ICF category title Example Classification

e110 Products or substances for personal consumption "[...] currently, for example, 11 months ago let's say I started to live a normal life, thanks to cortisone and immunoglobulin. And now, when I get up I take cortisone and..." Facilitator

 “[…] what the doctors do is give you a medication that works well for you, because before getting to that point I tried 4; all of them were bad for me.” Barrier

e355 Health professionals “I went to the psychologist and he told me ‘Look, write. If it’s hard for you to talk to someone, then write’, and I had no problem with that. It also helped me a lot to get out everything I was holding inside, the feeling of being powerless and all that.” Facilitator

 Well, for that, for example, you go to the GP. In my case, yes, but what happens is that the GPs confuse you.” Barrier 

e580 Health services, systems and policies "[...] because of course, physiotherapy, you have to pay for it, and physiotherapy is very important because it helps a lot." Facilitator

 “but I do see […] that there is a lack of resources so that the appointments are not so long, the waiting time, etc.” Barrier

e310 Immediate family “When I go for a walk in the afternoon, I always go with my husband, he always comes with me, I always hold on to him.” Facilitator

 “My husband’s help? He’s used to me doing everything for him, always…” Barrier

e115 Products and technology for personal use in daily living "I sit on a stool, in the shower; I have everything adapted, because I couldn't even get in; handrails everywhere for the shower." Facilitator

 “…but these days, except this: I can’t open jars that are tightly closed.” Barrier

e150 Design, construction and building products and technology of buildings for public use “[…] if there’s an elevator, I take the elevator. If not, if there are stairs up, well I have to take it very calmly, because it’s hard. If not, I get fatigued.” Facilitator

 "I can't go up stairs, I can't walk." 

 Barrier

e340 Personal care providers and personal assistants "[...] the last two years, I'm getting a little better, less pain; before, I’m in pain 24 hours. When I’m sleepy, I can’t get up, for example. I might someone to hold on to. Facilitator

e225 Climate “because in summer it’s a happy time and you go out, etc.” Facilitator

 "[...] but in winter, you have to stay at home more, or go out less because of the rain or the cold, and I think that being like that, so still, it’s like the pain is more concentrated”. Barrier

e210 Physical geography “For example, when I go out for a walk and I go to the mountains, I’m another person. I feel much better. ” Facilitator

 "Yes, my problem was going uphill., I live on a hill, so what happened? Of course, I had a really hard time going up a whole way and I had to stop half way up the street more than once." Barrier

e535 Communication services, systems and policies “You have an enormous lack of knowledge and on the internet they at least explain something.” Facilitator

 "I think many things are missing; people working on these rare diseases. There is a lack of information." Barrier

e555 Associations and organizational services, systems and policies "[...] if they put me in a group like an association, those that support ill people, well I think that would be good….” Facilitator

 “We’ve looked for groups, we’ve looked with my wife, with my family, and you find groups. There’s an association for this cancer, for that cancer, for stroke, for this and that, but there is no association for dermatomyositis.” Barrier

e460 Societal attitudes "But no one believes you're sick. That's another thing that affects me at least, isn’t it? Barrier

e120 Products and technology for personal indoor and outdoor mobility and transportation “[…] I fell down on the stairs. I started to walk again, with a walker” Facilitator

e560 Media services, systems and policies “[…] I have always looked at the disease online (internet), I like to find out everything.” Facilitator

 “But, of course, I don’t know if the thing is good or bad. There is a lack of information about this.” Barrier

e590 Labor and employment services, systems and policies “I had to ask for it [a leave of absence from work]. I was almost a year away. And when I felt better I told the doctor: ‘I need to go back to my routine, because I have to see if I can or not’.” Facilitator

 “How is it possible that a person with so much knowledge has to go on unemployment or has to work doing…oh well, whatever.” Barrier

7. Please clarify what is meant by “certain problems related with the disability caused by the disease” in the discussion (page 9, paragraph 1).

[Response] We were referring to several problems related to the disease (eg, muscle weakness, arthritis, fatigue…). In order to clarify this, we have modified the sentence accordingly: “…and certain problems related with the disability caused by the disease such as muscle weakness, intolerance to physical activity, and fatigue, with their consequences on mobility.

8. In the discussion, the sentence “The use of patient-reported outcomes derived from the focus group strategy to raise awareness of the issues that matter to patients is an essential part that should be considered in future studies (26).” (page 10, paragraph 2) does not appear to be related to the rest of the paragraph which discusses the role of caregivers. Please rephrase or move this statement to its own paragraph.

[Response] We have rewritten this paragraph and joined it to the previous one as examples of how the focus group approach brought to light the patients’ needs, as follows:

In addition to the classical approach (drug therapy) for managing these patients, other elements such as adapted stools for the shower, banisters to help going upstairs, and adapted living spaces focusing on building designs, were identified as factors that contribute to aid everyday functioning. As systematic reviews of orthotic devices for patients with neuromuscular problems have failed to find high-quality evidence proving their effectiveness (24), there is a lack of reliable information for physicians and patients in this line. The recent emergence of what may be useful devices, such as exoskeletons for myositis patients (25), could be of help for the problems in category e115 (Products and technology for personal use and daily living) and e120 (Products and technology for personal indoor and outdoor mobility), brought to light by patients in the focus groups. The role of relatives or assistants was also significant, as they help with walking and standing up, and confer reassurance on the patients’ ability to walk by avoiding the risk of falls. Thus, the importance of family members as caregivers emerged in the sessions. The use of patient-reported outcomes derived from focus groups to raise awareness of issues that matter to patients is precisely what enabled identification of these facilitators. Hence, the focus group strategy is a worthwhile approach that should be considered in future studies (26). 

9. Please shorten the concluding paragraph and move new suggestions to the main discussion text.

[Response] Considering the comment of the reviewer, we have placed points from the last paragraph in appropriate places in the Discussion. 

Patients need more useful lay information and dedicated associations where solutions can be sought for their self-perceived problems, similar to those already functioning around the world (eg, The Myositis Association, in the United States) (e555 Associations and organizational services, systems, and policies). (p.11, last paragraph)

Contact e-mails for health professionals, webs (internet) (e560 Media Services, systems and policies), and handouts, fliers, or patient page leaflets focused on IIM should be developed to improve access to this information (27). Web meetings and telematic symposiums should be conducted to empower patients to gain knowledge about their disease, the available therapeutic options, and the environmental modifications that will improve their well-being and HRQoL (p. 12 first paragraph)

and we have left the concluding paragraph as follows: 

“In conclusion, the ICF categories related to environmental factors identified here are of value for clinicians treating patients with these rare diseases, as they raise awareness of the need to reinforce the facilitating role of specific environmental factors and reduce the negative impact of others to improve the patients’ daily functioning.”

Minor points

1. Please standardize in-line citation formatting throughout the manuscript (e.g., citation of reference 10 on page 4, paragraph 3).

[Response] Done.

2. Please remove the “a” in the following sentence “Sporadic inclusion body myositis was... and a to some extent…” (page 5, paragraph 3).

[Response] Done.

3. Please remove the opening sentence of the “Environmental factors” subsection of “Results” (page 8, paragraph 3) as the definition of environmental factors has already been provided in the “Introduction”.

[Response] Done.

4. Reference 23 (Opinc AH, Brzezińska OE, Makowska JS. Disability in idiopathic inflammatory myopathies: questionnaire-based study. Rheumatol Int. 2019; 39:1213-1220.) does not appear to be cited within the manuscript. Please clarify its role in the text.

[Response] Done.

5. Reference 27 (Punnoose AR, Burke AE, Golub RM. JAMA patient page. Muscular dystrophy. JAMA. 2011; 306:2526.) is a patient information sheet for “Muscular dystrophy”. Please use a more relevant reference (for example, the Patient Fact Sheet on inflammatory myopathies as published by the American College of Rheumatology at https://www.rheumatology.org/Portals/0/Files/Inflammatory-Myopathies-Fact-Sheet.pdf).

[Response] Done.

Reviewer #3: 

Well-written study that gives an insightful understanding into the challenges faced by a group of patients with an uncommon rheumatic disease (IIM). Excellent use of an organized ICF framework, focused group, and well-conducted qualitative/thematic analysis of the environmental factors (that act as facilitators or barriers for this group of patients) by 2-3 independent persons as coders of meaning units. The study fulfils all field of the publication criteria including good demonstration of rigorous statistical analysis in qualitative studies. Good discussion points, including a balanced recognition of important limitations (eg single centre & hence caution with generalisability of the results) and strengths. Your study inspires and gives a great model that other centres around the world (of different social and cultural make-up) can adopt to study the challenges that their patients with IIM/other rheumatic diseases face.

Minor grammatical errors (2 of which stood out to me) did not affect the overall good quality of this manuscript.

[Response] Thank you very much for your kind comments.

We have realized that there was a mistake in data in table III, we have amended it in the new version of the manuscript (in red). Sorry for the inconvenience.

Sincerely yours

Albert Selva-O’Callaghan

---

## [Decision Letter · Decision Letter 1]

21 Dec 2020

“Functioning in adult patients with idiopathic inflammatory myopathy: Exploring the role of environmental factors using focus groups”.

PONE-D-20-10513R1

Dear Dr. Selva-O'Callaghan

We’re pleased to inform you that your manuscript has been judged scientifically suitable for publication and will be formally accepted for publication once it meets all outstanding technical requirements.

Kind regards,

Mahmoud Abu-Shakra, MD

Academic Editor

PLOS ONE

Additional Editor Comments (optional):

Reviewers' comments:

Reviewer's Responses to Questions

**Comments to the Author**

1. If the authors have adequately addressed your comments raised in a previous round of review and you feel that this manuscript is now acceptable for publication, you may indicate that here to bypass the “Comments to the Author” section, enter your conflict of interest statement in the “Confidential to Editor” section, and submit your "Accept" recommendation.

Reviewer #3: All comments have been addressed

2. Is the manuscript technically sound, and do the data support the conclusions?

Reviewer #3: Yes

3. Has the statistical analysis been performed appropriately and rigorously? 

Reviewer #3: Yes

4. Have the authors made all data underlying the findings in their manuscript fully available?

Reviewer #3: Yes

5. Is the manuscript presented in an intelligible fashion and written in standard English?

Reviewer #3: Yes

6. Review Comments to the Author

Reviewer #3: Well done on addressing the reviewers' comments. The revised manuscript is better & more clearly written.

Minor points for correction:

(1) Please delete ", and" after (e5):

"The environmental factors reported most often were those related to products and technology (e1) support and relationships (e3), services, systems and policies (e5), and (Table III).

(2) Is reference (23) incorrectly referenced here in the 1st sentence, 2nd paragraph of the "Discussion" section?:

Patients with idiopathic inflammatory myopathies can be highly dependent because of muscle weakness, joint inflammation, or the principal organ involvement, affecting, for example, respiration or the swallowing process (23).

7. PLOS authors have the option to publish the peer review history of their article (what does this mean?). If published, this will include your full peer review and any attached files.

Reviewer #3: No

---

## [Editor Report · Acceptance letter]

21 Jan 2021

PONE-D-20-10513R1 

Functioning in adult patients with idiopathic inflammatory myopathy: Exploring the role of environmental factors using focus groups 

Dear Dr. Selva-O'Callaghan:

I'm pleased to inform you that your manuscript has been deemed suitable for publication in PLOS ONE. Congratulations! Your manuscript is now with our production department. 

Kind regards, 

on behalf of

Dr. Mahmoud Abu-Shakra 

Academic Editor

PLOS ONE